# Influence of Sagittal Lumbopelvic Morphotypes on the Range of Motion of Human Lumbar Spine: An In Vitro Cadaveric Study

**DOI:** 10.3390/bioengineering9050224

**Published:** 2022-05-20

**Authors:** Wei Wang, Chao Kong, Fumin Pan, Wei Wang, Xueqing Wu, Baoqing Pei, Shibao Lu

**Affiliations:** 1Department of Orthopedics, Xuanwu Hospital, Capital Medical University, Beijing 100053, China; wangwei37@buaa.edu.cn (W.W.); kong988500@163.com (C.K.); fumin.pan@outlook.com (F.P.); 13910604946@163.com (W.W.); 2National Clinical Research Center for Geriatric Diseases, Beijing 100053, China; 3Beijing Key Laboratory for Design and Evaluation Technology of Advanced Implantable & Interventional Medical Devices, Beijing Advanced Innovation Center for Biomedical Engineering, School of Biological Science and Medical Engineering, Beihang University, Beijing 100083, China; xueqingwu@buaa.edu.cn

**Keywords:** sagittal parameters, range of motion, human lumbar spine, correlation analysis, in vitro experiment

## Abstract

Background: Although spinopelvic radiographs analysis is the standard for a pathological diagnosis, it cannot explain the activities of the spine in daily life. This study investigates the correlation between sagittal parameters and spinal range of motion (ROM) to find morphological parameters with kinetic implications. Methods: Six L1–S1 human lumbar specimens were tested with a robotic testing device. Eight sagittal parameters were measured in the three-dimensional model. Pure moments were applied to simulate the physiological activities in daily life. Results: The correlation between sagittal parameters and the ROM was moderate in flexion and extension, but weak in lateral bending and rotation. In flexion–extension, the ROM was moderately correlated with SS and LL. SS was the only parameter correlated with the ROM under all loading conditions. The intervertebral rotation distribution showed that the maximal ROM frequently occurred at the L5–S1 segment. The minimal ROM often appeared near the apex point of the lumbar. Conclusion: Sagittal alignment mainly affected the ROM of the lumbar in flexion and extension. SS and apex may have had kinetic significance. Our findings suggest that the effect of sagittal parameters on lumbar ROM is important information for assessing spinal activity.

## 1. Introduction

The transition from a C-shaped to an S-shaped spine has led to humans being the only vertebrates in the world that evolved to walk upright. Lumbar lordosis is unique to humans and not found in other species. The sagittal S-shaped curve of the human spine minimizes the energy expenditure of the back muscles while maintaining balance and stability [1]. Sagittal alignment of the spine is a recent and booming concept for understanding and treating spinal pathology.

Over the past 15 years, epidemiological and clinical studies have demonstrated that these sagittal parameters vary in a range of normality and are correlated with each other to maintain proper alignment in healthy subjects [2,3]. Roussouly et al. proposed four types of sagittal alignment of the normal spine that significantly differ from each other [4]. Multiple studies demonstrated the importance of sagittal balance in developing therapeutic strategies for many spinal disorders [5,6,7]. Therefore, the analysis of full spinopelvic radiographs is the standard for providing information on pathological diagnosis or preoperative planning in clinics [5,8].

Several spinopelvic sagittal parameters were described as realignment goals in patients with various spinal disorders [9,10]. While previous analytical studies mainly focused on imaging parameter analysis of the spinopelvic sagittal alignment [11,12], it is rare to see reports on the relationship between variations in the normal alignment and range of motion (ROM) of the spine. Indeed, standard radiographs are produced in a constrained position and restrained environment. In these conditions, how the radiographic spinal alignment affects the activities of the spine in daily life it cannot be explained, and the validity of this parameter assessment is questioned.

Therefore, this study preliminarily investigates whether different sagittal alignment morphotypes have various kinetic characteristics based on in vitro biomechanical tests of the cadaveric spine. The relationship between the sagittal parameters and the ROM of the spine was analyzed under different daily loading scenarios to find real morphological parameters with the kinetic implications. These results partially address this lack of basic knowledge of whether and how various sagittal alignments affect the activity of the spine in daily life.

## 2. Materials and Methods

### 2.1. Specimen Preparation

The experimental scheme in this study was approved by the biological and medical ethics committee of Beihang University (no.: BM20190009). Six lumbar specimens (L1–S1, 3 males, 3 females, mean age 46.7 years, range 32–64) were employed from a human donor spine. To ensure healthy conditions of the lumbar specimen, a history of back surgery, bony defects, disc degeneration, tumors, scoliosis, or prolonged bed rest before death was excluded from this study. Spiral computed tomography (CT) with a slice thickness of 0.6 mm (Light Speed Pro16, GE, Waukesha, WI, USA) was used to measure the sagittal parameters of each specimen in the next step. The specimen was carefully dissected to remove soft tissues while preserving intervertebral discs, ligaments, and facet joints [13]. The upper surface of vertebra L1 and the caudal end of S1 were embedded in polymethylmethacrylate (PMMA) and mounted in custom containers of the testing device. All specimens were wrapped in cling film to minimize water loss, and stored at −20 °C [14].

### 2.2. Sagittal Parameter Measurement

Eight sagittal parameters of the lumbar–pelvic specimen were measured in this study, namely, pelvic incidence (PI), sacral slope (SS), pelvic tilt (PT), lumbar lordosis (LL), the apex of the lordosis (Apex), upper arc, the lumbar title angle (LTA), and the number of vertebrae in the lordosis (NVL), as shown in Figure 1. The definition of each parameter was based on the study published by Duval-Beaupere and Roussouly [4,15].

### 2.3. Testing Devices

In this study, a robotic testing device (NX100MH6, Kabushiki-gaisha Yasukawa Denki, Kitakyushu, Japan) was performed that had been previously published to measure the force-displacement behavior of lumbar segments [14], as shown in Figure 2. A force-moment sensor (Gamma, ATI Industrial Automation, Apex, NC, USA) was attached to the head of the robotic arm to record forces and moments, and provide feedback. The L1 vertebra was also attached to the head of the robotic arm next to the sensor, and the S1 vertebra was fixed to the base frame. A three-dimensional opt-electric camera system (Optotrak Certus, Northern Digital Inc., Waterloo, ON, Canada) was applied to capture the rotation and displacement of the lumbar by recording the position of a set of markers. The default algorithm of the photoelectric camera system was used to determine the three-dimensional coordinates of the center of the S1 and principal directions. Five markers were fixed on L1, L2, L3, L4, and L5. 

### 2.4. Testing Protocol

A pure load control protocol was used for testing at a constant loading rate of 1.0°/s [13,16]. Specimens were tested under six pure moments, including 7.5 Nm in flexion and extension, 7.5 Nm in lateral bending, and 5 Nm in axial rotation [17,18]. The optimal loading path was determined by the robotic system at a 10% increment of the target load (7.5/5 Nm), and 4.5 loading cycles were applied to the specimen. The first 1.5 was used as the front cycle to minimize the viscoelastic effect, and the last 3 cycles were for subsequent analysis. Specimens were kept moist with 0.9% saline during the testing processing.

### 2.5. Data Analysis

Data were analyzed using SPSS software (IBM Corp, Armonk, NY, USA). The sagittal parameter and the intersegmental ROM of each lumbar specimen were measured and analyzed in all loading conditions. Spearman’s correlations were used to analyze the relationships between sagittal parameters and the ROM under all the loading conditions (using correlation coefficient and significance values r and *p*). Correlations were assumed to be strong (r = 0.80–1.00), moderate (r = 0.50–0.79), weak (r = 0.20–0.49), or no correlation (r < 0.20). A significant correlation was defined when *p* < 0.05. 

## 3. Results

### 3.1. Specimen Sagittal Parameters 

The lumbopelvic parameters of the six specimens are shown in Table 1. The average values for PI, PT, and SS were 46.88°, 10.67° and 36.22°, respectively. The average value for LL was 49.37°. The number of lumbar vertebrae is 5.01, that is, the inflection point of the spine from kyphosis to lordosis was near the thoracolumbar junction (T12–L1). The apex of lumbar lordosis appeared, on average, below the center of L4, from proximal upper L3 to the upper endplate of S1. The upper arc was the difference between LL and SS, with an average value of 14.90°. LTA averaged −4.75°, with a range from −5.2° to −5.9°.

### 3.2. Intervertebral Rotations

Under different loading conditions, the torque–displacement curves of the six specimens in vitro were hysteresis circles, as shown in Figure 3. The ROM was described as the average of the six values on the upper and lower boundaries of the hysteresis curve. Under the same loading conditions, the whole ROM of the L1–S1 segments varied between lumbar specimens due to structural and organizational differences. In general, the maximal ROM of the L1–S1 segments occurred in flexion ranging from 15.32° to 23.46°, and the minimum appeared in axial rotation ranging from 4.72° to 9.44°. The ROM varied between 9.37° and 22.18° MPa, and 10.50° and 20.46° under extension and lateral bending, respectively (Figure 4). The standard deviation of measurement was between 0.23° and 0.88°, and the error was within 5%.

### 3.3. Correlation Analysis

Correlation between sagittal parameters of lumbopelvic and intervertebral rotations under different loading conditions is shown in Table 2. Correlation between sagittal parameters and the ROM was moderate under flexion and extension loading, but weak under lateral bending and axial rotation loading. Under flexion and extension loading, the ROM was moderately correlated with SS (r = 0.63 and r = 0.53) and LL (r = 0.51 and r = 0.67), while the Upper Arc (r = 0.42 and r = 0.41), Apex (r = 0.33 and r = 0.36) and PI (r = 0.36 and r = 0.34) had weak effect on the ROM. Under the lateral bending and axial rotation loadings, the ROM was weakly correlated with the SS, with the r ranging from 0.28 to 0.36. There was no correlation between Roussouly’s classification [4,11] of the specimens and the ROM under all loading conditions.

### 3.4. Intervertebral Rotation Distribution

The distribution of the intervertebral rotation for L1–L2, L2–L3, L3–L4, L4–L5, and L5–S1 segments was ranged in the six specimens under different loading conditions, as shown in Figure 5. Under flexion, extension, and lateral bending loading, the maximal ROM appeared at the L5–S1 segment. The minimal ROM occurred at the L3–L5 segments near the apex point. The ROM of the lumbar from L1 to S1 segments decreased first and then increased, presenting a smile curve. Under the axial rotation loading, the ROM of each vertebra varied and fluctuated in six lumbar specimens. In addition, for the Type 1 specimen (1), the L5–S1 segment accounted for a larger proportion of the overall ROM of the lumbar spine under flexion and extension loads (Figure 6). Under the same loading condition, the ROM distribution of Type 2 specimens (2 and 3) showed a certain gradient, while that of Types 3 and 4 (4–6) was evenly distributed in the lumbar spine specimens. Under left–right bending loading, different types of specimens showed similar trends in the ROM distribution. Under the axial rotation loading, the ROM distribution varied between specimens. 

## 4. Discussion

As the loss of functional ROM is one of the main components leading to spinal disease, ROM analysis may also serve as a tool for patient identification and subsequent diagnostic purposes. The sagittal alignment of the spine is a recently developed concept to understand the mechanical equilibrium mechanism and the geometric characteristics of pathological deformity of the spine. Although clinical studies show that the functional ROM of the spine is determined by its structural morphology, the interrelation between the lumbar ROM and spinopelvic parameters remains unclear. In this in vitro biomechanical study utilizing human lumbar specimens, we preliminarily investigated the relationship between sagittal parameters and lumbar ROM under different daily loading scenarios. The correlation between the sagittal parameters and lumbar ROM was strong in flexion and extension, moderate in lateral bending, but weak in axial rotation. According to the classical Roussouly’s sagittal classification, the intervertebral rotation distribution for L1–L2, L2–L3, L3–L4, L4–L5, and L5–S1 segments was in a range in different types of normal lumbar sagittal alignment.

The present study reported functional ROM of L5–S1 segments in sagittal alignment ranging from 15.32° to 23.46° in flexion and 9.37° and 22.18°, which was in agreement with the 21.5 ± 7.3° in flexion and 9.3 ± 3.2° in extension reported by Rohlmann et al. [19]. Renner et al. [20] also reported that the normal functional ROM of L1–S1 segments was 49.7 ± 9.7° under pure-moment loading (8 Nm, flexion; 6 Nm, extension). Our measured ROM fell within the higher range of its measurements. The sagittal parameters of the human lumbar spine specimens in this study were mainly derived from normal Chinese adults, and the average of PI, PT, SS, and LL were less than those of the previously studied Caucasians by Roussouly et al. [4,21]. Roussouly et al. [4] reported that PI, SS, PT, and LL were 51.91 ± 10.71°,11.99 ± 6.46°, 39.92 ± 8.17°, and 61.43 ± 9.72°, respectively. Our measurements are consistent with those published by Endo et al. [22] in Japanese youth, with PI PT, LL, and SS values of 46.7 ± 8.9°, 13.2 ± 8.2°, 34.6 ± 7.8°, and 43.4 ± 14.6°, respectively. Our study reproduced the experimental results in both lumbar sagittal parameters and ROM to a good extent, although it is difficult to make direct comparisons with other studies due to inconsistent structural parameters and simulated loading conditions. 

Sagittal alignment plays a critical role in keeping the biomechanical adaptation and compensation of the spine and is the main influencing factor for obtaining an economic physiological position and motion. In this study, sagittal parameters had a moderate and weak effect on the ROM response of the lumbar spine under sagittal loading conditions (in flexion and extension) and lateral bending, respectively, but almost no effect under axial rotation loading. During the flexion and extension loading, the ROM was moderately correlated with SS and LL, and weakly correlated with Upper Arc, PI, and Apex. However, there was only a weak correlation between the ROM and SS in lateral bending and axial rotation. It is worth noting that SS is the only sagittal parameter associated with ROM under all loading conditions. That suggested that SS had the possibility of reflecting lumbar motion capacity, and explained its biomechanical significance as a decisive parameter for Roussouly sagittal classification.

The intervertebral rotation distribution showed that the maximal ROM frequently occurred at the L5–S1 segment under the flexion, extension, and lateral bending loading. Coherently with clinical evidence, the L5–S1 segments are at higher risk for disc degeneration and herniation. Gay et al. [23] found that degenerative grade had a significant effect on dynamic range of motion with differences between normal discs and higher grade degenerative discs. The minimal ROM often occurred at the L3–L5 segments near the Apex point. The specific anatomic turning point of the Apex may determine the stability center of the relative motion of the lumbar spine. Sebaaly et al. [6] noted that restoring the sagittal apex of lumbar lordosis helped reduce the incidence of proximal connection kyphosis from 41.4% to 13.5%. On the basis of Rousslouly’s sagittal classification of the lumbar, the L5–S1 segment of the Type 1 specimen accounted for a proportion of the overall ROM of the lumbar spine under flexion and extension loadings, while Type 3 and 4 models with a good lordosis shape had a more uniform rotation distribution at each motor function segment. In clinical cases, Type 1 patients frequently developed disc degeneration and herniation at the L5–S1 level. Roussouly and Adams et al. [11,24] also suggested that Type 1 patients were prone to early disc degeneration and herniation.

Our findings showed a relationship between sagittal parameters and lumbar ROM under different daily loading scenarios, and led to the hypothesizing about the kinetic significance of SS and apex. However, there are several limitations to the current study. First, due to the lack of in vitro lumbar specimens and the difficulty in obtaining CT data of the cadaver spine, the number of specimens that met the experimental requirements was limited. Second, due to the complexity of the in vitro experiment process and the customization of the loading device, our test results cannot be directly compared with other experimental results. Third, the mechanical loading of specimens in this experiment did not take into account the influence of muscles and other soft tissues, and the results were somewhat different from the real state of the lumbar spine in the body. Despite these limitations, our study can provide insights into how lumbar morphology influences lumbar motion and better understanding of the kinetic implications of sagittal parameters. 

## 5. Conclusions

Our study demonstrated that sagittal spinopelvic alignment mainly affected the motion response of the lumbar under sagittal loading conditions (in flexion and extension), and SS and LL were the main factors correlated to the ROM. The effect of sagittal parameters on the lumbar ROM is important information for assessing the function and stability of the spine. Further study should focus on the relationship between ROM and changes in sagittal plane alignment in patients with degenerative spinal diseases.

## Figures and Tables

**Figure 1 bioengineering-09-00224-f001:**
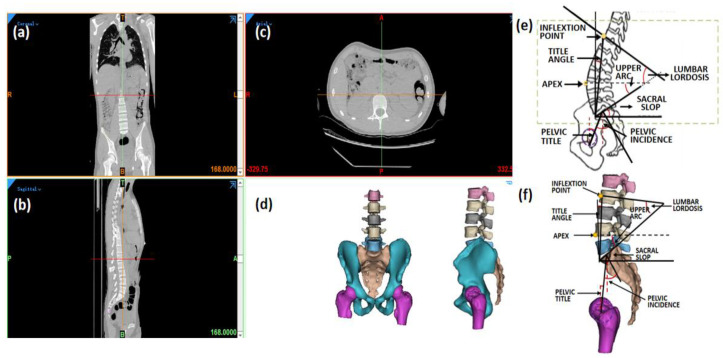
Measurement of sagittal parameters in 3D lumbopelvic model. (**a**) Coronal view, (**b**) sagittal view, and (**c**) axial view of CT images; (**d**) 3D lumbopelvic model; (**e**) definition of lumbopelvic parameter; (**f**) measurement in 3D lumbopelvic model.

**Figure 2 bioengineering-09-00224-f002:**
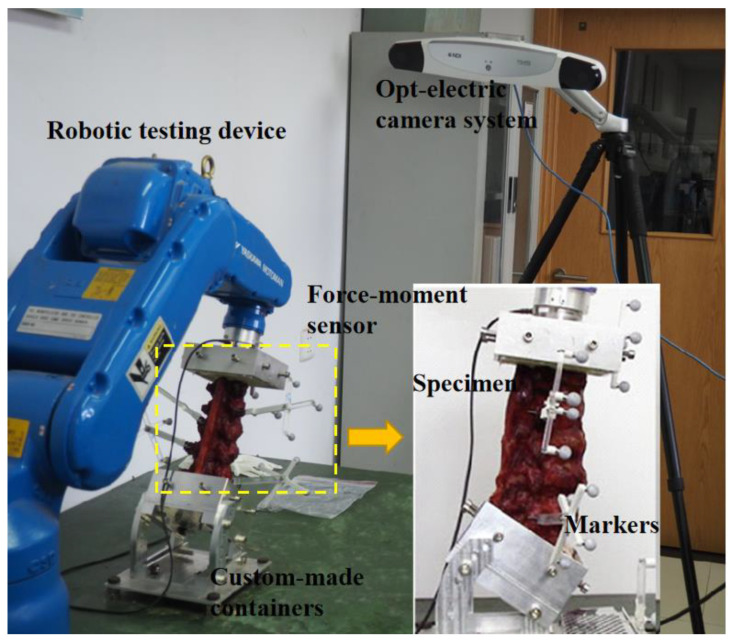
Image of robotic testing device with embedded specimen.

**Figure 3 bioengineering-09-00224-f003:**
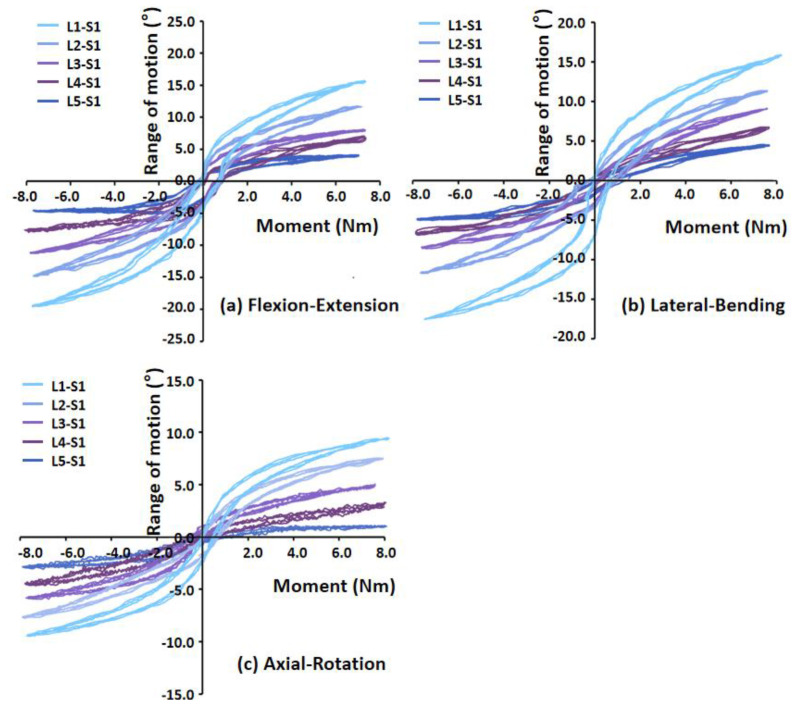
Torque–displacement curves of Specimen 2 in (**a**) flexion−extension (7 Nm), (**b**) lateral bending (7 Nm), (**c**) axial rotation (5 Nm).

**Figure 4 bioengineering-09-00224-f004:**
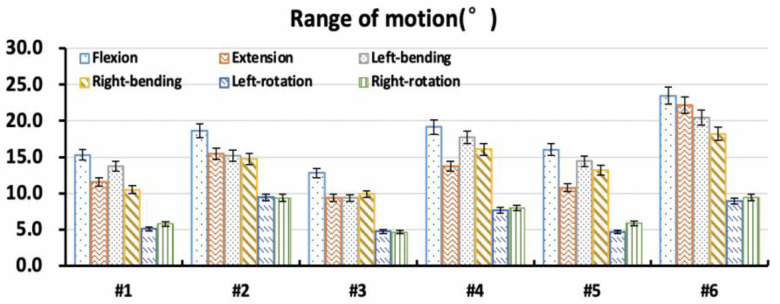
Overall range of motion of specimens in vitro.

**Figure 5 bioengineering-09-00224-f005:**
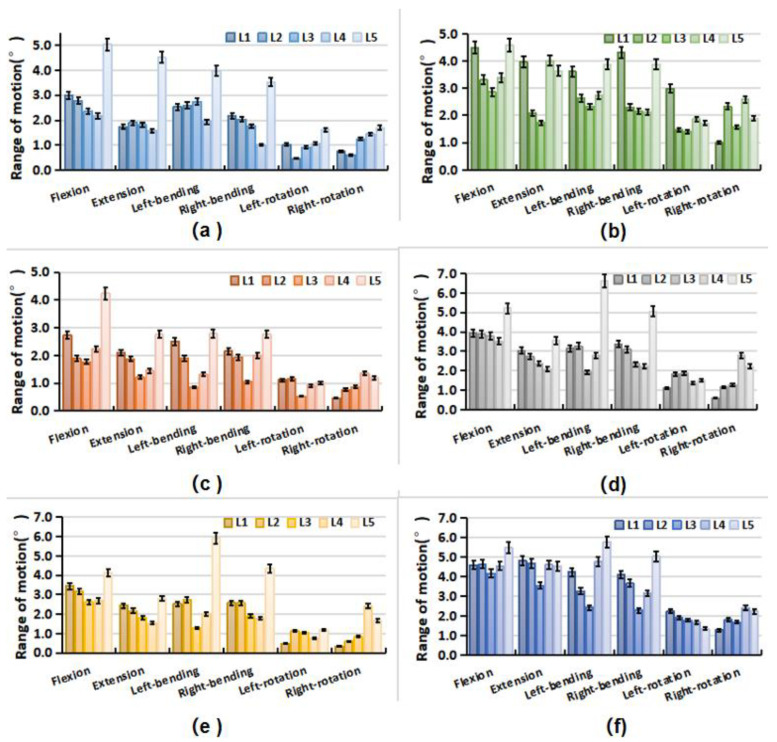
Range of motion of each vertebra in lumbar specimens under different loading. (**a**) Specimen-1; (**b**) Specimen-2; (**c**) Specimen-3; (**d**) Specimen-4; (**e**) Specimen-5; (**f**) Specimen-6.

**Figure 6 bioengineering-09-00224-f006:**
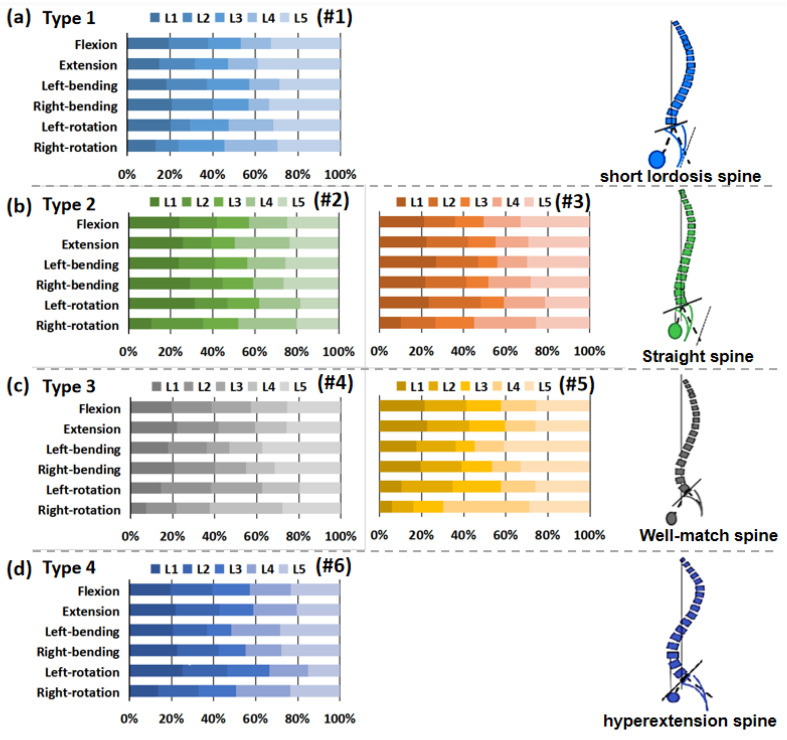
Percentage of range of motion of each vertebra in lumbar specimens under different loading. (**a**) Type 1 (1); (**b**) Type 2 (2 and 3); (**c**) Type 3 (4 and 5); (**d**) Type 4 (6).

**Table 1 bioengineering-09-00224-t001:** Sagittal parameters of lumbar–pelvic specimens.

Number	1	2	3	4	5	6	Mean ± SD
Sex	Female	Male	Male	Female	Male	Female	--
PI	36.9	39.6	44.2	47.5	54.1	59.0	46.88 ± 7.74
PT	9.4	10.2	11.4	10.1	10.7	12.2	10.67 ± 0.92
SS	27.5	29.4	32.8	37.4	43.4	46.8	36.22 ± 7.06
LL	40.3	43.2	48.2	52.4	53.9	58.2	49.37 ± 6.18
Apex	Upper L5	Base L4	Base L4	Middle L4	Middle L4	Base L3	--
Upper arc	14.0	13.9	15.4	14.8	14.7	16.6	14.90 ± 0.91
LTA	−5.2	−4.4	−4.2	−5.7	−5.9	−3.07	−4.75 ± 0.97
NVL	4.3	4.6	4.9	5.0	4.8	5.0	4.77 ± 0.25
Type	Type 1	Type 2	Type 2	Type 3	Type 3	Type 4	--

**Table 2 bioengineering-09-00224-t002:** Correlation between lumbar–pelvic parameter and range of motion.

	Type	PI (°)	SS (°)	PT (°)	LL (°)	Apex	Upper Arc (°)	LTA (°)	NVL
Flexion	0.28	0.36 *	0.63 * 7	−0.15	0.51 * 4	0.33 *	0.42 *	−0.14	−0.06
Extension	0.19	0.34 *	0.53 *	−0.04	0.67 *	0.36 *	0.41 *	0.17	0.15
Left-bending	0.10	−0.171	**0.32** *****	−0.29	−0.20	0.16	0.24	−0.20	−0.26
Right-bending	0.24	0.14	**0.35** *****	0.15	−0.27	0.24	0.20	0.30 *****	0.27
Left-rotation	0.13	−0.06	**0.36** *****	−0.08	−0.20	0.21	−0.07	0.24	0.21
Right-rotation	−0.04	0.19	**0.28** *****	0.12	−0.07	0.13	−0.06	0.13	−0.11

* Significant difference *p* < 0.05.

## Data Availability

Not applicable.

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
