# Peer review of "Influence of Sagittal Lumbopelvic Morphotypes on the Range of Motion of Human Lumbar Spine: An In Vitro Cadaveric Study"

_bioengineering, 2022, doi:10.3390/bioengineering9050224_

Round 1

Reviewer 1 Report

Dear Authors, "Influence of sagittal lumbopelvic morphotypes on the range of  motion of human lumbar spine: an in vitro study" is a well written paper with a good investigation about the correlation between sagittal parameters and spinal range of motion  to find out morphological parameters with kinetic implications. It will be for sure useful for the scientific community. 

Author Response

Reviewer #1: Dear Authors, "Influence of sagittal lumbopelvic morphotypes on the range of motion of human lumbar spine: an in vitro study" is a well written paper with a good investigation about the correlation between sagittal parameters and spinal range of motion to find out morphological parameters with kinetic implications. It will be for sure useful for the scientific community..

Response: We greatly appreciate the reviewer’s insightful comments that help us improve the quality of our manuscript. We have checked English language and style throughout the paper.

Thank you again for your hard work and valuable time. We are glad that our work has been recognized.

Best wishes.

Authors

Reviewer 2 Report

This is an in vitro cadaveric study focusing on the influence of sagittal lumbopelvic morphotypes on the range of motion of human lumbar spine. Overall, this is an interesting study and the preparation of this manuscript is good. I do not have major concerns. Here, I have a few minor concerns as below.

  1. Title should be modified a bit. This is an in-vitro cadaveric study.
  2. Abstract
  • What does “pure moments” mean?
  • All abbreviations need to be defined when used the first time.
  1. Some language errors have been found. Please check and correct those errors throughout this manuscript.
  2. In the section of materials and methods, “To ensure healthy conditions of the lumbar specimen, a history of back surgery, bony defects, disc degeneration, tumors, scoliosis, or prolonged bed rest before death were excluded from this study”. Whether did the donors suffer from any diseases that would impact the motion of spine before? Such as ankylosing spondylitis? Are any radiographs or MRI images of spine from these donors available?
  3. Please refer to this paper (Gay et al., Clin Biomech (Bristol, Avon), 21(9):914-9. doi: 10.1016/j.clinbiomech.2006.04.009), and discuss more.
  4. The section of conclusions is currently wordy and redundant, needing to be simplified. Please make the conclusions more comprehensive and concise.

Author Response

We greatly appreciate the reviewer’s insightful comments and suggestions, that help us improve the quality of the manuscript. We have responded to these comments and suggestions in a point-by-point fashion.
